# Multi-Wavelength Excitable Multicolor Upconversion and Ratiometric Luminescence Thermometry of Yb^3+^/Er^3+^ Co-Doped NaYGeO_4_ Microcrystals

**DOI:** 10.3390/molecules29204887

**Published:** 2024-10-15

**Authors:** Hui Zeng, Yangbo Wang, Xiaoyi Zhang, Xiangbing Bu, Zongyi Liu, Huaiyong Li

**Affiliations:** School of Materials Science and Engineering, Laboratory of Sensitive Materials and Devices Shandong Department of Education, Liaocheng University, Liaocheng 252059, China; z932141607@163.com (H.Z.); zx20221234@163.com (X.Z.); bxbgo300@163.com (X.B.); a1801488112@163.com (Z.L.)

**Keywords:** lanthanide upconversion, controllable luminescence, ratiometric thermometry, Er^3+^ sensitization

## Abstract

Excitation wavelength controllable lanthanide upconversion allows for real-time manipulation of luminescent color in a composition-fixed material, which has been proven to be conducive to a variety of applications, such as optical anti-counterfeiting and information security. However, current available materials highly rely on the elaborate core–shell structure in order to ensure efficient excitation-dependent energy transfer routes. Herein, multicolor upconversion luminescence in response to both near-infrared I and near-infrared II (NIR-I and NIR-II) excitations is realized in a novel but simple NaYGeO_4_:Yb^3+^/Er^3+^ phosphor. The remarkably enhanced red emission ratio under 1532 nm excitation, compared with that under 980 nm excitation, could be attributed to the Yb^3+^-mediated cross-relaxation energy transfers. Moreover, multi-wavelength excitable temperature-dependent (295–823 K) upconversion luminescence realizes a ratiometric thermometry relying on the thermally coupled levels (TCLs) of Er^3+^. Detailed investigations demonstrate that changing excitation wavelength makes little difference for the performances of TCL-based ratiometric thermometry of NaYGeO_4_:Yb^3+^/Er^3+^. These findings gain more insights to manipulate cross-relaxations for excitation controllable upconversion in single activator doped materials and benefit the cognition of the effect of excitation wavelength on ratiometric luminescence thermometry.

## 1. Introduction

Lanthanide-activated upconversion luminescence materials extend the excitation wavelength from the traditional ultraviolet and visible region to the near-infrared (NIR) region, enabling the full utilization of the advantages of near-infrared light sources, such as low background fluorescence interference and greater biological tissue penetration ability [1,2]. The characteristic multi-peak emissions of lanthanide ions due to their abundant energy levels and electronic transitions endow lanthanide-activated materials with multicolor upconversion luminescence, allowing the broad applications in optical display, anti-counterfeiting, and information security [3]. Multicolor upconversion luminescence has usually been achieved by managing the composition of lanthanide-doped materials, for example, the types of hosts, the types and concentrations of sensitizers and activators, and the distribution of doped ions [4]. Besides this, excitation wavelength controllable multicolor upconversion luminescence offers an in situ and reversible luminescence management without the need of changing the material composition [5,6]. However, for most of the reported upconversion materials, they could only produce a luminescence of specific color when excited at a specific wavelength. Even the available excitation wavelength is added by introducing new sensitizers and energy transfer paths [7]; the emission color usually stays the same due to the same luminous ions.

Great attempts have been made to realize excitation wavelength controlled multicolor upconversion, and exciting results have been achieved. By arranging different activators and sensitizers in separate areas in multi-shell nanostructures, previous research has demonstrated a versatile excitation wavelength-dependent colorful upconversion luminescence, which is orthogonal upconversion [8]. For example, elaborately synthesized multi-shell NaGdF_4_:Yb/Tm@NaGdF_4_@NaYbF_4_:Nd@Na(Yb,Gd)F_4_:Ho@NaGdF_4_ nanoparticles emit green light under 808 nm excitation while blue light under 980 nm excitation [9]; NaGdF_4_:Yb/Er@NaYF_4_:Yb@NaGdF_4_:Yb/Nd@NaYF_4_@NaGdF_4_:Yb/Tm@NaYF_4_ nanoparticles exhibit prominently blue and green luminescence with 980 and 796 nm excitation, respectively [10]. Orthogonal upconversion could also be achieved by the assembly of different nanoparticles capable of producing different luminescent colors at different wavelengths of excitation [11,12]. Moreover, by managing the energy migration processes in multi-shell nanostructures with well-designed sensitizer distribution, Zhang’s group acquired switchable colorful upconversion luminescence from red to green in single activator Er^3+^-doped nanoparticles by changing the excitation wavelength from 980 to 808 nm [13,14]. Although great progress has been made in excitation wavelength controllable multicolor upconversion, it is always necessary to construct complex core–shell or cluster structure to reduce cross-relaxation energy loss and avoid interference between excitation-dependent energy transfer paths. Therefore, it is highly valuable and desirable to develop simple material available for excitation wavelength controlled multicolor upconversion. As well known, Er^3+^ ions can not only produce tunable luminescence with multiple bands covering visible and NIR regions but can also be excited by several wavelengths of NIR light, such as 808, 980, and 1532 nm, owing to their abundant equidistant energy levels [15,16]. Meanwhile, co-doping Er^3+^ with other lanthanide ions (Tm^3+^, Ho^3+^, Yb^3+^, etc.) [17,18,19,20] or transition metal ions (Mn^2+^, Fe^3+^, etc.) [21,22] could introduce special energy transfer processes to realize multicolor luminescence. So, Er^3+^ is an appropriate dopant to serve simultaneously as an activator and sensitizer to achieve excitation wavelength-dependent multicolor upconversion in simple single activator doped materials.

Sodium yttrium germanate (NaYGeO_4_) has been recently investigated as a promising optical matrix material for efficient lanthanide and bismuth luminescence because of its multiple cation sites available for accommodating various luminous ions and a wide bandgap up to 6 eV [23,24,25,26]. Meanwhile, NaYGeO_4_ was reported to have a relatively low phonon energy of about 800 cm^−1^ [27,28], which suggests its potential to host lanthanide ions for efficient upconversion luminescence. However, present research on lanthanide-doped NaYGeO_4_ all focus on the conventional Stokes luminescence [29,30]. Therefore, herein, we synthesized Yb^3+^/Er^3+^ co-doped NaYGeO_4_ phosphors and explored their upconversion luminescence properties systematically. When excited by traditional near-infrared I (NIR-I) light of 980 nm, NaYGeO_4_:Yb^3+^/Er^3+^ phosphors emit bright green light through the typical energy transfer upconversion (ETU) processes of the Yb^3+^/Er^3+^ ion pair. Switching the excitation source to near-infrared II (NIR-II) light of 1532 nm leads to the excited state absorption (ESA) upconversion luminescence of Er^3+^, where a distinct yellow or orange–yellow emission could be observed due to the significantly increased red emission ratio. Mechanistic studies combining the steady-state and transient spectroscopic analyses reveal that the bidirectional cross-relaxation energy transfers between Er^3+^ and Yb^3+^ are responsible for the enhanced red emission under 1532 nm excitation. Moreover, ratiometric luminescence thermometry is achieved in a temperature range as wide as 295–823 K under multiple excitation wavelengths by using the luminescence intensity ratio (LIR) of the thermally coupled levels (TCLs, ^2^H_11/2_, and ^4^S_3/2_) of Er^3+^. Results demonstrate the excellent sensitivity, uncertainty, and repeatability of ratiometric thermometry based on TCLs and the little effect of changing excitation wavelength on thermometric performances.

## 2. Results and Discussion

### 2.1. Structure, Composition and Morphology

NaYGeO_4_ owns an orthorhombic olivine structure with a *Pnma* space group. Three kinds of cationic polyhedra, GeO_4_ tetrahedra, NaO_6_ and YO_6_ octahedra, connect to each other by sharing a corner or edge (Figure 1a). Detailed structure information can be found in a crystallographic information file from the Inorganic Crystal Structure Database, which has been included in the Appendix A. Due to the same valence and the very close ionic radii between Yb^3+^/Er^3+^ and Y^3+^ (1.008/1.030 Å for Yb^3+^/Er^3+^ and 1.040 Å for Y^3+^, coordination number = 6) [31], Yb^3+^ and Er^3+^ ions can be easily doped into the NaYGeO_4_ lattice through substituting Y^3+^ to realize upconversion luminescence. The obtained XRD patterns of synthesized NaYGeO_4_:xYb^3+^/2%Er^3+^ (x = 2–48%) are in good agreement with the standard patterns of orthorhombic NaYGeO_4_ (PDF#88-1177), as in Figure 1b. Rietveld refinement of obtained XRD patterns was performed to analyze the subtle change in crystallographic structure with the increase in Yb^3+^ concentration. The resulting refinement patterns and structural parameters are presented in Appendix A. Low and stable refinement factors (*R*_wp_, *R*_p_, and χ^2^) indicate that the obtained refinement results are reliable. Samples with 2–48% Yb^3+^ doping maintain the orthorhombic phase with the *Pnma* space group. Calculated cell volume decreases continuously from 384.7716 to 378.8438 Å^3^ with the increase of Yb^3+^ concentration (Figure 1c), demonstrating the isomorphic substitution of Y^3+^ ions by smaller Yb^3+^ ions. The XPS result of the exemplary NaYGeO_4_:18%Yb^3+^/2%Er^3+^ in Figure 1d reveals distinctly the existence of Na 1*s*, Y 3*d*, Ge 2*p*, and O 1*s* orbitals. Specifically, the high-resolution result of Yb 4*d* (183.6 eV) and Er 4*d* (168.0 eV) orbitals proves the successful incorporation of Yb^3+^/Er^3+^ in NaYGeO_4_ (the inset in Figure 1d).

The SEM image in Figure 1e shows that NaYGeO_4_:18%Yb^3+^/2%Er^3+^ particles are irregular lumpy microcrystals in the size of about several microns. The morphology was not significantly changed when adjusting Yb^3+^ concentration (Appendix A). The EDS spectrum also demonstrates the remarkable presence of Na, Y, Ge, O, and Yb elements, while no obvious peak of Er could be discerned due to the low doping concentration (Figure 1f). Based on the detected atomic percentages by the EDS spectrum, the elemental composition of Na:Y:RE:O for NaYGeO_4_:18%Yb^3+^/2%Er^3+^ sample could be calculated to 0.92:1.20:1:4.40, which is close to the theoretical ratio of 1:1:1:4. As the Yb^3+^ doping concentration increases, atomic percentages of Na, Ge, O, and Er remain almost unchanged, while those of Y and Yb go evidently opposite trends (Appendix A). Therefore, the detected Yb concentration increases linearly from 2.5% to 55.9% when increasing Yb^3+^ doping concentration from 2% to 48%, as shown in Appendix A. Although the detected concentration of Yb^3+^ is not quite the same as the feed concentration, the trends are consistent. The elemental mapping images in Figure 1g show the even distribution of Na, Y, Ge, O, Yb, and Er elements in NaYGeO_4_:18%Yb^3+^/2%Er^3+^.

### 2.2. Upconversion Properties

As expected, NaYGeO_4_:Yb^3+^/Er^3+^ produces the typical upconversion luminescence of Er^3+^ under excitations of both 980 nm NIR-I and 1532 nm NIR-II light. Figure 2a shows the upconversion spectra of NaYGeO_4_:18%Yb^3+^/2%Er^3+^ under 980 and 1532 nm excitations. Three main emission bands at 515–570, 634–705, and 783–826 nm, respectively, are from the ^2^H_11/2_/^4^S_3/2_ → ^4^I_15/2_, ^4^F_9/2_ → ^4^I_15/2_, and ^4^I_9/2_ → ^4^I_15/2_ energy level transitions of Er^3+^. It is noticeable that 980 nm excitation gives rise to comparable intensity for green (515–570 nm) and red (634–705 nm) emissions, while 1532 nm excitation leads to red emission much stronger than green. The red/green ratios relying on the integral intensity are calculated to 1.1 and 9.0 for 980 and 1532 nm excitations, respectively. This implies a different luminescence color under different excitation conditions.

Three main emission bands enhance first and then weaken with the rising of Yb^3+^ doping concentration under both 980 and 1532 nm excitations (Figure 2b,c). The highest upconversion emission intensity is obtained at 18% Yb^3+^ doping for 980 nm excitation, while 8% Yb^3+^ doping gives the highest emission for 1532 nm excitation. Higher Yb^3+^ doping brings about obvious concentration quenching effects under both excitation wavelengths. The influence of Er^3+^ concentration on luminescence was also considered. Variation of Er^3+^ concentration has little effect on the luminescence intensity under 980 nm excitation, and the highest intensity is achieved at 2% Er^3+^ doping (Appendix A). This is because heavy activator content may result in cross-relaxation energy loss rather than more emissions. Increasing Er^3+^ concentration continuously enhances 1532 nm excited upconversion luminescence due to the increasing absorption for excitation light (Appendix A). This is because 1532 nm excitation leads to the self-sensitization luminescence of Er^3+^, which can be found in the following luminescence mechanisms. Taking into account the stronger luminescence under 980 nm excitation and larger red/green ratio under 1532 nm excitation, the concentration of Er^3+^ was chosen as 2%. As presented in Figure 2d, the red/green ratio increases gradually and slightly from 0.7 to 2.0 with the increased Yb^3+^ doping from 2% to 48% under 980 nm excitation, while it increases drastically from 3.6 to 11.9 for 2–28% Yb^3+^ doping and then decreases to 6.4 under 1532 nm excitation. Apparently, the red/green ratios obtained by 980 nm excitation are lower than those obtained by 1532 nm excitation at all Yb^3+^ doping concentrations. The biggest difference in red/green ratios (1.6 and 11.9 for 980 and 1532 nm excitations, respectively) is achieved at 28% Yb^3+^ doping. Accordingly, the upconversion luminescence color of NaYGeO_4_:xYb^3+^/2%Er^3+^ (x = 2–48%) changes from green to yellow or orange–yellow when switching the excitation from 980 nm NIR-I to 1532 nm NIR-II light, as explicitly shown in Figure 2e. This excitation wavelength-dependent multicolor luminescence is indicative of a potential of this material for luminescent anti-counterfeiting application.

We compared the upconversion spectra of samples with and without Yb^3+^ doping. Under 980 nm excitation, NaYGeO_4_:2%Er^3+^ shows absolutely negligible emissions compared with the 18% Yb^3+^ co-doped counterpart, as given in Figure 3a. This corresponds to the much lower absorption cross-section of Er^3+^ (~10^−21^ cm^2^) than Yb^3+^ (~10^−20^ cm^2^) for 980 nm excitation light [32]. So, the ETU mechanism is undoubtedly the dominant mechanism for 980 nm excited upconversion luminescence of NaYGeO_4_:Yb^3+^/Er^3+^. For the upconversion spectra under 1532 nm excitation, Figure 3b displays noticeable emissions of NaYGeO_4_:2%Er^3+^ with comparable intensity to NaYGeO_4_:18%Yb^3+^/2%Er^3+^. This is because 1532 nm photons could only be absorbed by Er^3+^ while Yb^3+^ has no matched energy levels [3]. Nonetheless, the upconversion red/green ratio of NaYGeO_4_:2%Er^3+^ (calculated as 2.2) is significantly lower than that of NaYGeO_4_:18%Yb^3+^/2%Er^3+^ (calculated as 9) and those of samples with 2–48% Yb^3+^ doping (calculated as 3.6–11.9). This implies the key role of Yb^3+^ in the red predominant upconversion luminescence under 1532 nm excitation.

To further sketch out the energy processes for the upconversion luminescence under both excitation wavelengths, the excitation photon numbers (*n*) required for main emissions were then figured out by analyzing the double logarithmic relationships between emission intensity (*I*) and excitation power (*P*). According to the equation $I\propto P^{n}$, the value of *n* can be determined as the slope of the linear fit of the double logarithmic relationship [33]. From the results in Appendix A, under 980 nm excitation, the *n* values for three main emissions at 515–570, 634–705, and 783–826 nm are calculated near to 2. This indicates that 980 nm excitation mainly leads to two-photon upconversion luminescence. Based on the results in Appendix A, three main emissions under 1532 nm excitation could be determined as three-photon predominant upconversion processes.

The upconversion luminescence mechanisms are schematically illustrated in Figure 3c,d. The luminescence of NaYGeO_4_:Yb^3+^/Er^3+^ under 980 nm excitation involves the typical ETU upcoversion processes as shown in Figure 3c. Yb^3+^ ions absorb the energy of 980 nm photons and are excited, then, two successive energy transfer processes from Yb^3+^ to Er^3+^ excite the later to ^4^F_7/2_ level. Er^3+^ ions at ^4^F_7/2_ level may reach ^2^H_11/2_, ^4^S_3/2_, ^4^F_9/2_, and ^4^I_9/2_ levels by several nonradiative relaxations. The second-step energy transfer may also excite Er^3+^ from the ^4^I_13/2_ to ^4^F_9/2_ level. Finally, for Er^3+^ ions at ^2^H_11/2_, ^4^S_3/2_, ^4^F_9/2_, and ^4^I_9/2_ levels, radiative transitions to the ground state level take place and corresponding emission bands around 532, 558, 660, and 812 nm are generated, respectively. Figure 3d describes the self-sensitization luminescence mechanism of Er^3+^ under 1532 nm excitation. 1532 nm photons are absorbed by Er^3+^ ions due to their suitable equidistant energy levels. Ground state absorption (GSA) of ^4^I_15/2_ → ^4^I_13/2_ and the subsequent ESA process of ^4^I_13/2_ → ^4^I_9/2_ populates Er^3+^ ions at the ^4^I_9/2_ level. Er^3+^ ions are further excited to the ^2^H_11/2_ level through the following ESA process of ^4^I_9/2_ → ^2^H_11/2_, and then may also relax nonradiatively to ^4^S_3/2_, ^4^F_9/2_, and ^4^I_9/2_ levels [34,35]. Finally, radiative transitions from ^2^H_11/2_, ^4^S_3/2_, ^4^F_9/2_, and ^4^I_9/2_ levels to the ground state level produce corresponding emissions as marked.

To the distinctly enhanced proportion of red luminescence after Yb^3+^ doping under 1532 nm excitation, following cross-relaxation energy transfer processes may be reasonable (Figure 3d): ET1, ^4^I_11/2_ (Er^3+^) + ^2^F_7/2_ (Yb^3+^) → ^4^I_15/2_ (Er^3+^) + ^2^F_5/2_ (Yb^3+^); ET2, ^4^S_3/2_ (Er^3+^) + ^2^F_7/2_ (Yb^3+^) → ^4^I_13/2_ (Er^3+^) + ^2^F_5/2_ (Yb^3+^); ET3, ^4^I_13/2_ (Er^3+^) + ^2^F_5/2_ (Yb^3+^) → ^4^F_9/2_ (Er^3+^) + ^2^F_7/2_ (Yb^3+^). Energy transfer from Er^3+^ to Yb^3+^ is a prerequisite for Yb^3+^ participating in the energy processes and subsequently affecting the populations of Er^3+^ at green and red emission levels. ET1 is likely to occur due to the highly resonant energy levels involved [36,37]. ET2 may also happen with the assistance of lattice phonon [38,39]. These energy transfer processes from Er^3+^ to Yb^3+^ could be verified by the gradually shortened lifetime of the green emission level ^4^S_3/2_ from 15.5 to 2.0 μs as increasing Yb^3+^ concentrations (Figure 3e), because these processes could suppress the population of Er^3+^ at ^4^S_3/2_ level directly (ET2) or indirectly (ET1, which deactivates ^4^I_11/2_ level thus affects the population of the higher level ^4^S_3/2_). The lifetime of the red emission level ^4^F_9/2_ should also be shortened due to the energy transfer from Er^3+^ to Yb^3+^, but there is a distinctly different trend at low Yb^3+^ concentrations of 0–8% (Figure 3f). This incipient upward trend indicates a further energy transfer from Yb^3+^ to Er^3+^, promoting the population of Er^3+^ at red emission level ^4^F_9/2_. So, ET3 probability takes place because Er^3+^ ions are prone to reach ^4^I_13/2_ levels by GSA processes [40]. The ultimate shortening in the lifetimes of red emission level ^4^F_9/2_ could be attributed to the suppressed energy transfer from Yb^3+^ to Er^3+^, which is due to the excitation energy loss caused by the concentration quenching effect at high Yb^3+^ concentrations. On the whole, these cross-relaxation energy transfer processes between Er^3+^ and Yb^3+^ shall boost the population of Er^3+^ ions at the ^4^F_9/2_ level and, thus, increase the red emission ratio.

### 2.3. Ratiometric Temperature Sensing Using Thermally Coupled Levels

The change in upconversion spectra with temperature suggests that NaYGeO_4_:Yb^3+^/Er^3+^ can be applied for ratiometric luminescent thermometry. As shown in Figure 4a,d and Appendix A, the upconversion luminescence intensity at different bands of NaYGeO_4_:18%Yb^3+^/2%Er^3+^ evidently reduces as increasing temperature from 295 to 823 K, excepting the noteless variation of 515–541 nm emission. Different change rates for different emission bands allow for ratiometric luminescent temperature sensing. We explored the ratiometric temperature sensing based on the well-known TCLs ^2^H_11/2_ and ^4^S_3/2_ of Er^3+^. The upconversion spectra in the green region normalized to 558 nm in Figure 4b,e distinctly show the gradually increased ratio for 515–541 nm emission (from ^2^H_11/2_ → ^4^I_15/2_ transitions of Er^3+^) relative to 541–575 nm emission (from ^4^S_3/2_ → ^4^I_15/2_ transition of Er^3+^), when increasing temperature from 295 to 823 K under both 980 and 1532 nm excitation. This could be ascribed to the thermally promoted population of Er^3+^ ions at the ^2^H_11/2_ level from ^4^S_3/2_ level following the Boltzmann distribution [41,42].

The corresponding LIR relying on the TCLs ^2^H_11/2_ and ^4^S_3/2_ at different temperature, *T*, can be usually determined by:(1)$$\mathrm{LIR}=\frac{I_{532}}{I_{558}}=C\;\exp(-\frac{\Delta E}{k_{B}T})$$
where *I*_532_ and *I*_558_ represent the integral emission intensity at 515–541 and 541–575 nm, respectively; *C* is a constant; Δ*E* is the energy gap between two emission levels ^2^H_11/2_ and ^4^S_3/2_; *k*_B_ is the Boltzmann constant (1.3806 × 10^−23^ J/K) [43,44,45]. Furthermore, Equation (1) can be mathematically transformed into Equation (2):(2)$$\mathrm{Ln}\;\mathrm{LIR}=\mathrm{Ln}C-\frac{\Delta E}{k_{B}T}$$
which reflects a simple linear relationship between Ln LIR and 1/*T*. Dependences of Ln LIR (*I*_532_/*I*_558_) against 1/*T* in Figure 4c,f are perfectly linear relationships, being fitted well by Equation 2 with high fitting coefficients of 0.9977 and 0.9990 for 980 and 1532 nm excitation, respectively. Detailed fitting equations have been included in Figure 4c,f, and it is clear that different excitation wavelengths give highly similar mathematical relationships between LIR (*I*_532_/*I*_558_) and temperature. The Δ*E* values between ^2^H_11/2_ and ^4^S_3/2_ levels were calculated as 709.6 and 713.6 cm^−1^ for 980 and 1532 nm excitation, respectively. These values agree with those usually reported in previous works of about 600–900 cm^−1^ [46]. Above analyses indicate that LIRs of TCLs under both 980 and 1532 nm excitation show excellent mathematical relationships with temperature in a wide range of 295–823 K, which suggests a promising application of synthesized NaYGeO_4_:Yb^3+^/Er^3+^ for temperature sensing.

In order to evaluate the thermometric sensitivity, the absolute sensitivity (*S*_a_) and relative sensitivity (*S*_r_), reflecting the change rate of the parameter LIR with temperature, were quantified using the following equations [47]:(3)$$S_{a}=\left|\frac{\mathrm{dLIR}}{dT}\right|=\;\mathrm{LIR}\;\frac{\Delta E\;}{k_{B}T^{2}}$$
(4)$$S_{r}=\frac{1}{\mathrm{LIR}}\cdot \left|\frac{\mathrm{dLIR}}{dT}\right|100\%=\frac{\Delta E\;}{k_{B}T^{2}}\;100\%$$

For LIR (*I*_532_/*I*_558_) of NaYGeO_4_:18%Yb^3+^/2%Er^3+^ under both 980 and 1532 nm excitation, obtained *S*_a_ values first increase and then decrease, while *S*_r_ values decrease continuously (Figure 5a,d). The optimal *S*_r_ are 1.17% and 1.18% K^−1^ for 980 and 1532 nm excitation, respectively. These are two very close results owing to the highly similar change trends in LIR (*I*_532_/*I*_558_) with temperature under 980 and 1532 nm excitation, as presented above in Figure 4c,f. Another important performance parameter of a thermometric method is the temperature uncertainty (*δT*), which was calculated by the following equation:(5)$$\delta T=\frac{1}{S_{r}}\frac{\delta \mathrm{LIR}}{\mathrm{LIR}}$$
where *δLIR/LIR* is the relative uncertainty of obtained LIR [48]. The value of *δLIR/LIR* was determined by several measurements at 298 K. The calculated *δT* increases continuously with temperature increasing under both 980 and 1532 nm excitation (Figure 5b,e), and the minimal values are 0.20 and 0.14 K, respectively. Moreover, periodic tests of two heating–cooling cycles in Figure 5c,f show excellent repeatability of LIR (*I*_532_/*I*_558_) at a range of selected temperatures, implying a reliable ratiometric temperature sensing method. These results, as summarized in Table 1, demonstrate a little effect of changing excitation wavelength on the performance of TCL-based thermometry, which is because different excitation wavelengths mainly affect the population routes of green and red emission levels rather than TCLs, as discussed in the above luminescent mechanism section.

## 3. Materials and Methods

### 3.1. Materials

Reagents including sodium carbonate (Na_2_CO_3_, 99.5%), yttrium oxide (Y_2_O_3_, 99.99%), germanium oxide (GeO_2_, 99.99%), ytterbium oxide (Yb_2_O_3_, 99.99%), and erbium oxide (Er_2_O_3_, 99.99%), were all purchased from Shanghai Aladdin Biochemical Technology Co., Ltd., (Shanghai, China). These raw materials without any purification were directly used for synthesis.

### 3.2. Synthesis

NaYGeO_4_:Yb^3+^/Er^3+^ samples were synthesized by the high-temperature solid-state method. Dopants Yb^3+^ and Er^3+^ were intendedly introduced to replace Y^3+^, and all the doping concentrations are molar percentages in this work. In a typical process, raw materials including Na_2_CO_3_, Y_2_O_3_, GeO_2_, Yb_2_O_3_, and Er_2_O_3_, were weighed accurately according to the stoichiometry ratio. The mixture of raw materials was fully ground with an agate mortar, and then transferred into an aluminum oxide crucible and placed in a furnace at 800 ℃ in atmosphere for 12 h pre-sintering. The obtained powder was ground again and sintered at 1200 ℃ in atmosphere for 12 h. All the heating and cooling rates were 5 ℃/min. After cooling to room temperature, the resulting sample was ground again for 10 min and collected for later use.

### 3.3. Characterization

Powder X-ray diffraction patterns of obtained samples were acquired on a D8 Advance diffractometer (Bruker, Karlsruhe, Germany) with Cu Kα radiation as the incident beam, with 2θ range of 10–75°, step angle of 0.01°, and dwell time of 0.2 s. X-ray photoelectron spectroscopy (XPS) results were collected on an XPS Microprobe (Thermo SCIENTIFIC ESCALAB Xi+, Waltham, MA, USA). Morphology and the energy-dispersive X-ray spectroscopy (EDS) characterization were conducted on a Field Emission Scanning Electron Microscope (Zeiss Merlin, Oberkochen, Germany). Upconversion luminescence spectra were measured on a FLS1000 fluorescent spectrometer (Edinburgh, Livingston, UK) with CW 980 and 1532 nm diode lasers (Changchun New Industries Optoelectronics Technology Co., Ltd., Changchun, China) as excitation sources. The upconversion luminescence spectra at 295–823 K were collected on the same spectrometer equipped with a HCP621G gas-tight thermal plate (Instec, Bergen, Norway).

## 4. Conclusions

In summary, excitation controlled multicolor upconversion has been realized in NaYGeO_4_:Yb^3+^/Er^3+^ phosphors. The traditional 980 nm NIR-I excitation produces bright green luminescence, whereas 1532 nm NIR-II excitation leads to distinct yellow/orange–yellow emission. Mechanistic analyses revealed Yb^3+^ induced cross-relaxation energy transfers that accelerate the generation of Er^3+^ at red emission level under 1532 nm excitation. Temperature-dependent upconversion luminescence studies established a ratiometric thermometry relying on the TCLs (^2^H_11/2_ and ^4^S_3/2_) of Er^3+^, under multi-wavelength excitations of 980 and 1532 nm. It was found that switching excitation wavelength has a very small effect on the performance of TCL-based thermometry. Finally, high relative sensitivity up to 1.18% K^−1^, low uncertainty as low as 0.14 K, and a wide working temperature range of 295–823 K were achieved. These results provide a novel but simple material, NaYGeO_4_:Yb^3+^/Er^3+^, allowing not only excitation wavelength controlled multicolor upconversion luminescence in a fixed material but also excellent ratiometric luminescence thermometry.

## Figures and Tables

**Figure 1 molecules-29-04887-f001:**
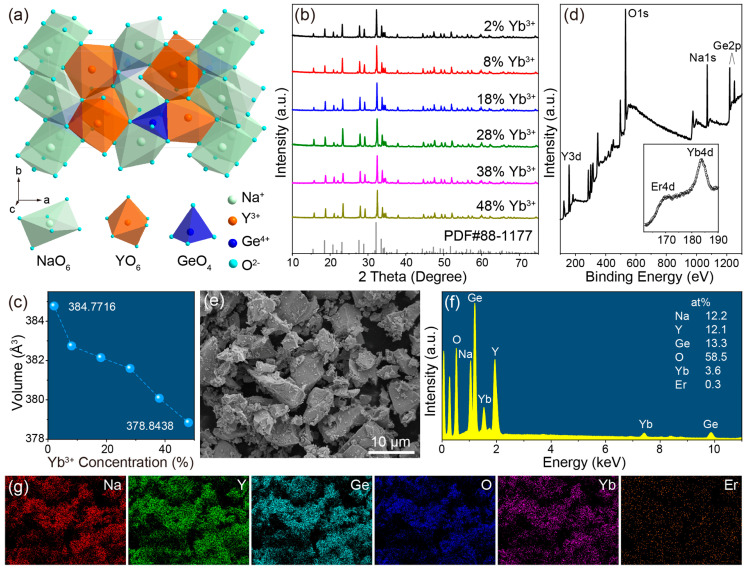
(**a**) Crystal structure of NaYGeO_4_ and the coordination polyhedra of NaO_6_, YO_6_, and GeO_4_. (**b**) XRD patterns of NaYGeO_4_:xYb^3+^/2%Er^3+^ microcrystals, x = 2–48%. The bar-like diffraction patterns at the bottom represent the standard data of orthorhombic NaYGeO_4_ (PDF#88–1177). (**c**) Unit cell volume calculated from Rietveld refinement results as a function of Yb^3+^ doping concentration. (**d**) XPS spectrum, (**e**) SEM image, and (**f**) EDS spectrum of NaYGeO_4_:18%Yb^3+^/2%Er^3+^. The inset in (**d**) is the high-resolution XPS spectrum in the range of 165–190 eV. (**g**) Elemental mappings corresponding to the SEM image in (**e**).

**Figure 2 molecules-29-04887-f002:**
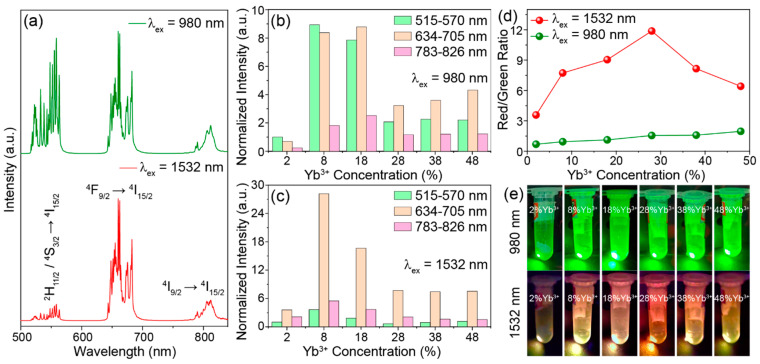
(**a**) Upconversion emission spectra of NaYGeO_4_:18%Yb^3+^/2%Er^3+^, under 980 and 1532 nm laser excitation. The integral intensity evolutions of 515–570, 634–705, and 783–826 nm emissions for NaYGeO_4_: xYb^3+^/2%Er^3+^ at increased Yb^3+^ concentrations, under (**b**) 980 and (**c**) 1532 nm excitation. (**d**) The upconversion red/green ratios and (**e**) luminescence photographs of NaYGeO_4_:xYb^3+^/2%Er^3+^ at increased Yb^3+^ concentrations under 980 and 1532 nm excitation.

**Figure 3 molecules-29-04887-f003:**
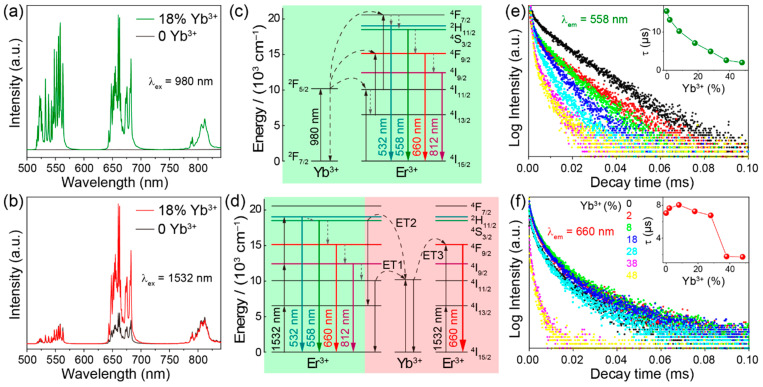
Upconversion emission spectra of NaYGeO_4_:18%Yb^3+^/2%Er^3+^ and NaYGeO_4_:2%Er^3+^, under (**a**) 980 and (**b**) 1532 nm excitation. Schematic upconversion luminescence mechanisms of NaYGeO_4_:Yb^3+^/Er^3+^ with (**c**) 980 and (**d**) 1532 nm excitation. Decay curves of NaYGeO_4_: xYb^3+^/2%Er^3+^ under 1532 nm excitation at (**e**) 558 and (**f**) 660 nm emissions; the insets show the calculated lifetimes as a function of Yb^3+^ concentration.

**Figure 4 molecules-29-04887-f004:**
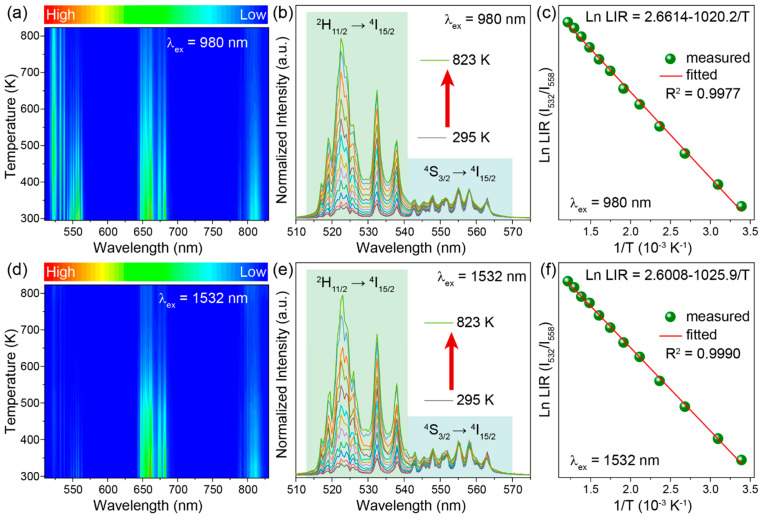
(**a**,**d**) Upconversion luminescence spectra, (**b**,**e**) normalized green upconversion spectra, and (**c**,**f**) calculated Ln LIR (*I*_532_/*I*_558_) of NaYGeO_4_:18%Yb^3+^/2%Er^3+^ in the temperature range of 295–823 K, under (**a**–**c**) 980 and (**d**–**f**) 1532 nm excitation.

**Figure 5 molecules-29-04887-f005:**
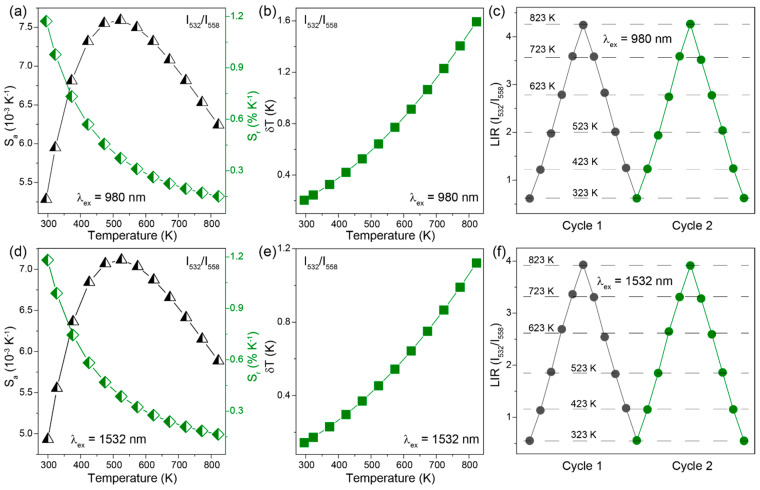
(**a**,**d**) Absolute sensitivity, *S_a_*, and relative sensitivity, *S_r_*. (**b**,**e**) Temperature uncertainty *δT* relying on LIR (*I*_532_/*I*_558_) of NaYGeO_4_:18%Yb^3+^/2%Er^3+^ at different temperatures. (**c**,**f**) LIR (*I*_532_/*I*_558_) at selected temperatures for two heating–cooling cycles between 323 and 823 K. Under (**a**–**c**) 980 and (**d**–**f**) 1532 nm excitation.

**Table 1 molecules-29-04887-t001:** Main parameters including the temperature range, *S_r_*, and *δT* of ratiometric thermometry using the upconversion luminescence from TCLs of NaYGeO_4_:18%Yb^3+^/2%Er^3+^.

Excitation Wavelength (nm)	LIR Used	Temperature Range (K)	*S_r_* (% K^−1^)	*δT* (K)
980	*I*_532_/*I*_558_ (TCLs)	295–823	0.15–1.17	0.20–1.59
1532	*I*_532_/*I*_558_ (TCLs)	295–823	0.15–1.18	0.14–1.12

## Data Availability

Data are available from authors upon reasonable requirement.

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
