# Peer review of "Multi-Wavelength Excitable Multicolor Upconversion and Ratiometric Luminescence Thermometry of Yb3+/Er3+ Co-Doped NaYGeO4 Microcrystals"

_molecules, 2024, doi:10.3390/molecules29204887_

Round 1
Reviewer 1 Report
Comments and Suggestions for Authors
In this article, the authors synthesized and described microcrystals of NaYGeO4:Yb3+/Er3+ in order to study the luminescent properties of this compound. They various the concentration of Yb in order to maximize the luminescence intensity. The article gives a lot of photophysical and structural data, and it is particularly interesting that the authors are exploring the potential of this material for luminescent thermometry. However, there are a few issues with the article that need to be addressed:
1) Section 2.1. Structure, composition and morphology: I would suggest to refine unit cell parameters and plot unit cell volume vs Yb3+ concentration to demonstrate isomorphic substitution Y3+ ions by Yb3+ ions
2) The amount of ytterbium varies in the samples, but the concentration of erbium is assumed to be fixed at 2%. The chosen Er3+ content needs to be justified.
3) Because the upconversion emission intensity and spectral shape depend on the excitation power (Fig S3, Supporting information), it is unreliable to use the Yb-Er materials as ratiometric temperature sensing using non-thermally coupled levels (Section 2.4). I would suggest to remove the section 2.4. from the article.
Author Response
Thank you very much for taking the time to review this manuscript. Please find the detailed responses below and the corresponding revisions highlighted in the re-submitted files.
In this article, the authors synthesized and described microcrystals of NaYGeO4:Yb3+/Er3+ in order to study the luminescent properties of this compound. They various the concentration of Yb in order to maximize the luminescence intensity. The article gives a lot of photophysical and structural data, and it is particularly interesting that the authors are exploring the potential of this material for luminescent thermometry. However, there are a few issues with the article that need to be addressed.
Comments 1: Section 2.1. Structure, composition and morphology: I would suggest to refine unit cell parameters and plot unit cell volume vs Yb3+ concentration to demonstrate isomorphic substitution Y3+ ions by Yb3+ ions.
Response 1: Thank you very much for the constructive suggestion. We have done Rietveld refinement and analyzed the change of unit cell volume. Obtained results are shown in Figures S1 and 1c, and Table S1. Corresponding description was highlighted in the first paragraph of Section 2.1 in the revised manuscript.
“Rietveld refinement of obtained XRD patterns was performed to analyze the subtle change of crystallographic structure with the increase of Yb3+ concentration. Resulted refinement patterns and structural parameters are presented in Figure S1 and Table S1. Low and stable refinement factors (Rwp, Rp, and c2) indicate that obtained refinement results are reliable. Samples with 2%-48% Yb3+ doping maintain the orthorhombic phase with Pnma space group. Calculated cell volume decreases continuously from 384.7716 to 378.8438 Å3 with the increase of Yb3+ concentration (Figure 1c), demonstrating the isomorphic substitution of Y3+ ions by smaller Yb3+ ions.”.
Comments 2: The amount of ytterbium varies in the samples, but the concentration of erbium is assumed to be fixed at 2%. The chosen Er3+ content needs to be justified.
Response 2: We have added the study on Er3+ concentration and corresponding description in the first paragraph of Page 5 as highlighted in the revised manuscript:
“The influence of Er3+ concentration on luminescence was also considered. Variation of Er3+ concentration has little effect on the luminescence intensity under 980 nm excitation, and the highest intensity is achieved at 2% Er3+ doping (Figure S4a,b). This is because heavy activator content may result in cross-relaxation energy loss rather than more emissions. Increasing Er3+ concentration continuously enhances 1532 nm excited upconversion luminescence due to the increasing absorption for excitation light (Figure S4c,d). This is because 1532 nm excitation leads to the self-sensitization luminescence of Er3+, which can be found in the following luminescence mechanisms. Taking into account the stronger luminescence under 980 nm excitation and larger red/green ratio under 1532 nm excitation, the concentration of Er3+ was chosen as 2%.”
Comments 3: Because the upconversion emission intensity and spectral shape depend on the excitation power (Fig S3, Supporting information), it is unreliable to use the Yb-Er materials as ratiometric temperature sensing using non-thermally coupled levels (Section 2.4). I would suggest to remove the section 2.4. from the article.
Response 3: Thank you very much for the insightful comment. We agree that using emissions from the non-thermally coupled green and red emission levels in Yb-Er materials for ratiometric thermometry is unreliable, and the obtained results also show poor dependence with Temperature. So, we have removed the section about ratiometric temperature sensing using non-thermally coupled levels in the revised manuscript.

Reviewer 2 Report
Comments and Suggestions for Authors
Manuscript demonstrates that only TCLs levels are the most promissing for thermometry, even at different excitation channels. Maybe it would be better prononced in manuscript more clearly.
Author Response
Thank you very much for taking the time to review this manuscript. Please find the detailed responses below and the corresponding revisions highlighted in the re-submitted files.
Comments: Manuscript demonstrates that only TCLs levels are the most promissing for thermometry, even at different excitation channels. Maybe it would be better prononced in manuscript more clearly.
Response: It is unreliable to use emissions from the non-thermally coupled green and red emission levels in Yb-Er materials for ratiometric thermometry, because the upconversion emission intensity ratio (such as red/green ratio) depends on the excitation power. The obtained results based on non-TCLs also show relatively poor dependences with Temperature. So, we have removed the section about ratiometric temperature sensing using non-TCLs in the revised manuscript, as suggested by Reviewer 1.
Reviewer 3 Report
Comments and Suggestions for Authors
This work investigates the multicolor upconversion luminescence of a novel NaYGeO4:Yb3+/Er3+ phosphor. The enhanced red emission rate is observed at 1532 nm excitation compared to 980 nm excitation, and the effect was attributed to the Yb3+ cross-relaxation energy transfers. The multiwavelength excitable temperature-dependent upconversion luminescence exhibits multimode ratiometric thermometry due to thermally and non-thermally coupled Er3+ levels. The upconversion luminescence was investigated and discussed for co-doped NaYGeO4 microcrystal synthesized and characterized using different techniques. The results indicate the new material is potentially excellent for ratiometric luminescence thermometry.
Yb3+/Er3+ co-doped NaYGeO4 phosphors were synthesized and explored for upconversion luminescence properties, and present potential for ratiometric luminescence thermometry. The manuscript presented a novel set of synthesized materials, which were extensively investigated and characterized, with potential for application in luminescence thermometry. Therefore, the work can be considered for publication in Molecules in its current form.
Author Response
Comments: This work investigates the multicolor upconversion luminescence of a novel NaYGeO4:Yb3+/Er3+ phosphor. The enhanced red emission rate is observed at 1532 nm excitation compared to 980 nm excitation, and the effect was attributed to the Yb3+ cross-relaxation energy transfers. The multiwavelength excitable temperature-dependent upconversion luminescence exhibits multimode ratiometric thermometry due to thermally and non-thermally coupled Er3+ levels. The upconversion luminescence was investigated and discussed for co-doped NaYGeO4 microcrystal synthesized and characterized using different techniques. The results indicate the new material is potentially excellent for ratiometric luminescence thermometry.
Yb3+/Er3+ co-doped NaYGeO4 phosphors were synthesized and explored for upconversion luminescence properties, and present potential for ratiometric luminescence thermometry. The manuscript presented a novel set of synthesized materials, which were extensively investigated and characterized, with potential for application in luminescence thermometry. Therefore, the work can be considered for publication in Molecules in its current form.
Response: Thank you very much for taking the time to review this manuscript and giving the positive comments
Reviewer 4 Report
Comments and Suggestions for Authors
In this work, the authors provide results for synthesized material with high relative sensitivity up to 374 1.18% K‒1, low uncertainty down to 0.14 K, and a wide working temperature range of 295-823K. The studied NaYGeO4:Yb3+/Er3+ phosphor is synthesized with increasing Yb content from 2% to 48% and observed changes in luminescent properties by two excitation wavelengths. The results achieved would contribute to the enrichment of structures suitable for applications in optics, anti-counterfeiting applications, and luminescence thermometry. The manuscript is relevant to the field and presented in a well-structured form. I have followed minor corrections which I believe will improve the manuscript:
1. The authors should include the XRD information in the following format:
- "Crystal Data for C42H63P (M = 598.89 g/mol): monoclinic, space group P21/c (no. 14), a = 33.5426(3) Å, b = 10.06131(11) Å, c = 46.2696(5) Å, β = 93.3735(9)°, V = 15588.1(3) Å3, Z = 16, T = 99.9(2) K, μ(CuKα) = 0.790 mm-1, Dcalc = 1.021 g/cm3, 98894 reflections measured (6.7° ≤ 2Θ ≤ 155.122°), 32038 unique (Rint = 0.0381, Rsigma = 0.0401) which were used in all calculations. The final R1 was 0.0562 (I > 2σ(I)) and wR2 was 0.1452 (all data)."
- Or include a suitable CIF file amongst the Supporting Materials.
2. The comment on lines 154 - 155 that the highest upconversion emission intensity obtained at 18% Yb3+ doping for 980 nm is difficult to confirm from Figure S2a because the intensity difference between samples with 8% and 18% Yb is not presented clearly while Figure 2c, b presents very good this observation. Please, choose a suitable reference to present of results.
Author Response
Thank you very much for taking the time to review this manuscript. Please find the detailed responses below and the corresponding revisions highlighted in the re-submitted files.
In this work, the authors provide results for synthesized material with high relative sensitivity up to 1.18% K‒1, low uncertainty down to 0.14 K, and a wide working temperature range of 295-823K. The studied NaYGeO4:Yb3+/Er3+ phosphor is synthesized with increasing Yb content from 2% to 48% and observed changes in luminescent properties by two excitation wavelengths. The results achieved would contribute to the enrichment of structures suitable for applications in optics, anti-counterfeiting applications, and luminescence thermometry. The manuscript is relevant to the field and presented in a well-structured form. I have followed minor corrections which I believe will improve the manuscript:
Response: Thank you very much for the positive comments.
Comments 1: The authors should include the XRD information in the following format:
"Crystal Data for C42H63P (M = 598.89 g/mol): monoclinic, space group P21/c (no. 14), a = 33.5426(3) Å, b = 10.06131(11) Å, c = 46.2696(5) Å, β = 93.3735(9)°, V = 15588.1(3) Å3, Z = 16, T = 99.9(2) K, μ(CuKα) = 0.790 mm-1, Dcalc = 1.021 g/cm3, 98894 reflections measured (6.7° ≤ 2Θ ≤ 155.122°), 32038 unique (Rint = 0.0381, Rsigma = 0.0401) which were used in all calculations. The final R1 was 0.0562 (I > 2σ(I)) and wR2 was 0.1452 (all data)."
Or include a suitable CIF file amongst the Supporting Materials.
Response 1: We have included the CIF file for NaYGeO4 amongst the Supporting Materials, and added the main crystallographic structural parameters from the Rietveld refinement of XRD patterns in Table S1 in the Supplementary Information.
Comments 2: The comment on lines 154-155 that the highest upconversion emission intensity obtained at 18% Yb3+ doping for 980 nm is difficult to confirm from Figure S2a because the intensity difference between samples with 8% and 18% Yb is not presented clearly while Figure 2c, b presents very good this observation. Please, choose a suitable reference to present of results.
Response 2: We have removed indistinguishable Figure S2 while maintained Figure 2b,c for clear presentation of these results.

Round 2
Reviewer 1 Report
Comments and Suggestions for Authors
Authors carefully revised manuscript and now it can be accepted